# Susceptibility Analysis of Geohazards in the Longmen Mountain Region after the Wenchuan Earthquake

**DOI:** 10.3390/ijerph19063229

**Published:** 2022-03-09

**Authors:** Shuai Li, Zhongyun Ni, Yinbing Zhao, Wei Hu, Zhenrui Long, Haiyu Ma, Guoli Zhou, Yuhao Luo, Chuntao Geng

**Affiliations:** 1College of Tourism and Urban-Rural Planning, Chengdu University of Technology, Chengdu 610059, China; shuailee.cn@gmail.com (S.L.); zhaoyinbing06@cdut.cn (Y.Z.); 2019020722@stu.cdut.edu.cn (W.H.); zhouguoli@stu.cdut.edu.cn (G.Z.); luoyuhao0723@gmail.com (Y.L.); gct199610@gmail.com (C.G.); 2College of Earth Sciences, Chengdu University of Technology, Chengdu 610059, China; 3School of Geography, Archaeology & Irish Studies, National University of Ireland, H91 CF50 Galway, Ireland; 4Human geography research center of Qinghai Tibet Plateau and its eastern margin, Chengdu University of Technology, Chengdu 610059, China; 5Sichuan Research Institute of Ecological Restoration of Land Space and Geohazard Prevention and Control, Sichuan Provincial Department of Natural Resources, Chengdu 610063, China; longzrei@gmail.com; 6College of Information, Shanghai Ocean University, Shanghai 201306, China; mahaiyu.cn@outlook.com

**Keywords:** geohazard susceptibility, superposition effect, random forest model, Longmenshan fault zone

## Abstract

Multitemporal geohazard susceptibility analysis can not only provide reliable results but can also help identify the differences in the mechanisms of different elements under different temporal and spatial backgrounds, so as to better accurately prevent and control geohazards. Here, we studied the 12 counties (cities) that were severely affected by the Wenchuan earthquake of 12 May 2008. Our study was divided into four time periods: 2008, 2009–2012, 2013, and 2014–2017. Common geohazards in the study area, such as landslides, collapses and debris flows, were taken into account. We constructed a geohazard susceptibility index evaluation system that included topography, geology, land cover, meteorology, hydrology, and human activities. Then we used a random forest model to study the changes in geohazard susceptibility during the Wenchuan earthquake, the following ten years, and its driving mechanisms. We had four main findings. (1) The susceptibility of geohazards from 2008 to 2017 gradually increased and their spatial distribution was significantly correlated with the main faults and rivers. (2) The Yingxiu-Beichuan Fault, the western section of the Jiangyou-Dujiangyan Fault, and the Minjiang and Fujiang rivers were highly susceptible to geohazards, and changes in geohazard susceptibility mainly occurred along the Pingwu-Qingchuan Fault, the eastern section of the Jiangyou-Dujiangyan Fault, and the riparian areas of the Mianyuan River, Zagunao River, Tongkou River, Baicao River, and other secondary rivers. (3) The relative contribution of topographic factors to geohazards in the four different periods was stable, geological factors slowly decreased, and meteorological and hydrological factors increased. In addition, the impact of land cover in 2008 was more significant than during other periods, and the impact of human activities had an upward trend from 2008 to 2017. (4) Elevation and slope had significant topographical effects, coupled with the geological environmental effects of engineering rock groups and faults, and river-derived effects, which resulted in a spatial aggregation of geohazard susceptibility. We attributed the dynamic changes in the areas that were highly susceptible to geohazards around the faults and rivers to the changes in the intensity of earthquakes and precipitation in different periods.

## 1. Introduction

Geohazards are geological effects or geological phenomena, which are mainly caused by geodynamic activities or abnormal changes in the geological environment [1]. With global climate change, frequent earthquakes, and intensified human activities, the frequency and intensity of geohazards around the world have been increasing [2]. The early assessment of regional geohazards mainly depends on field data, which is often time consuming to collect and is limited spatially and temporally. Remote sensing technology and geographic information systems not only provide efficient and quick data acquisition for large-scale geohazard investigation, but can also be used to monitor long-term deformation or displacement for landslides or debris flows [3]. Using the spectral, spatial, and morphological attributes and contextual information of geohazards in remote sensing images, researchers can quickly identify and investigate geohazards [4]. Different types of data, such as aerial images, satellite images, synthetic aperture radar, and lidar have been successfully applied to geohazard identification in different regions and even in mountainous areas [5]. Field surveys have evolved to be verification for the interpretation of aerial and spaceborne remote sensing rather than the primary method for data acquisition. To better prevent and control regional hazards and further reduce the harm of geohazards to life, property, and the ecological environment, geohazard assessment can be carried out after successfully obtaining and compiling data [6]. Based on the regional environmental background and related factors, geohazard susceptibility research mainly explores the spatial distribution of the characteristics of areas that are susceptible to geohazards and their driving mechanisms [7].

Geohazard susceptibility studies include both qualitative and quantitative methods [8]. Qualitative methods are based on the relevant aspects of geohazard investigation and determine the main impact factors of geohazard and delineate the areas susceptible to geohazards [9]. These qualitative methods include the Delphi method [10], analytic hierarchy process [11], weighted average method [12], and weighted linear combination [13]. Although the qualitative method is simple and easy to implement and takes into account the regional geological background and relevant historical information, it mainly relies upon expert experience, which can be highly subjective and have shortcomings [14]. Quantitative methods rely on different mathematical models to calculate the probability of geohazards on the basis of quantifying relevant influencing factors [15]. Early quantitative methods were based on statistical methods, such as the entropy index [16], weight of evidence [17], spatial autoregression [18], and geographically weighted regression [19], to describe the linear relationship between geohazards and impact factors, which can be less subjective [20]. In recent years, many new quantitative methods are gradually being applied in the field of geohazards, such as data mining, machine learning, and neural networks [21], which helps researchers break through the limitations of traditional statistical methods [22]. As a model capable of both classification and regression, random forest overcomes problems associated with complicated modeling processes and the weak interpretative ability of other machine learning algorithms [23], and it can avoid the high sensitivity problems of neural network algorithms and have a more stable model performance [24]. This algorithm has yielded good results in geohazard susceptibility studies in geomorphologically complex areas such as the Zagros Mountain in Iran [25], Woomyeon Mountain in Korea [26], East Sikkim in India [27], and the earthquake-prone regions of Hokkaido in Japan [28], Sicily in Italy [29], and Antalya in Turkey [30]. However, random forest requires high data accuracy, and overfitting may occur when using noisy data to build a model [31].

Located on the eastern edge of the Qinghai-Tibet Plateau, the Longmen Mountain region is an earthquake- and geohazard-prone area in China [32], with many active faults and complex topography and geological conditions [33]. The 8.0 Ms Wenchuan earthquake that occurred on 12 May 2008 triggered a large number of secondary geohazards, such as collapses, landslides, debris flows [34], and its damage to the stability of the mountain also affected the development of subsequent geohazards for a long time [35]. The 7.0 Ms Lushan earthquake that occurred on 20 April 2013, also had a certain impact on the Longmen Mountain region [36], but the heavy rainfall in the same year made the geohazard situation in the area more severe [37]. Studies have shown that strong earthquakes and heavy rainfall have increased or decreased the intensity of different inducing factors in different years [38], which in turn affects the dynamic changes of the spatial distribution of geohazard susceptibility. However, previous studies on the susceptibility of geohazards mostly used all geohazards in all or multiple years after the Wenchuan earthquake as parameters [39], and did not use special events as nodes for the division of time periods, which affected the tracking of geohazard susceptibility. Some studies have insufficiently considered the spatial heterogeneity of related factors, which has limited the accuracy of research in areas with complex geomorphology [40]. Moreover, some research results may underestimate the superimposition effect of earthquakes and precipitation and the amplification effect of precipitation on topographical factors, and may not fully evaluate the dynamic impact of earthquakes, rainfall, and other important geohazard inducing factors [41]. Some studies may not fully consider the impact of human activities and their derivative effects on geohazards in different time periods [42].

Based on previous research and after considering the changes in the intensity of earthquakes and precipitation, we used the year of the Wenchuan earthquake and the heavy rainfall event as the time node to divide our study into four periods: 2008, 2009–2012, 2013, and 2014–2017. We combined the geohazards after the Wenchuan earthquake and used the advanced random forest (RF) model to study the dynamic changes in geohazard susceptibility and its driving mechanism regarding the 12 May 2008, Wenchuan earthquake and the subsequent 10 years in the Longmen Mountain region. To improve the accuracy of identifying factors that influence geohazards in complex geomorphological areas, we performed secondary processing, such as continuity, grading, and weighting, on some factors. Our approach provides a new perspective for the study of long-term geohazards, and our data processing method could provide a reference for large-scale or complex geomorphological areas. Our results can support hazard prevention efforts in different scenarios and aid in the precise implementation of regional hazard prevention and control.

## 2. Materials and Methods

### 2.1. Study Area

The study area was in the transition zone between the Qinghai-Tibet Plateau and the Sichuan Basin (102°36′–105°38′ E, 30°44′–33°3′ N; Figure 1) and included Dujiangyan and 11 other counties (cities) with a total area of about 3.31 × 10^4^ km^2^. The elevation ranged from 400–6000 m, in a ladder shape from southwest to northeast. The Longmenshan fault zone in the area is located at the junction of the Yangtze block and the Songpan-Ganzi geosyncline fold system [43], with strong neotectonic movement, and develops several groups of main active faults, including the Jiangyou-Dujiangyan Fault (F1), the Yingxiu-Beichuan Fault (F2), the Pingwu-Qingchuan Fault (F3), and the Maoxian-Wenchuan Fault (F4). The exposed stratum in the study area is from Mesoproterozoic to Quaternary, and the lithology is mainly carbonate rock, clastic rock, and metamorphic rock [44]. This area is at the intersection of the mountainous subtropical humid monsoon climate zone and warm temperate continental semiarid monsoon climate zone, with a substantial gradient of climate. The precipitation in the study area is abundant, with a multi-year average precipitation of about 1050 mm, which is mainly concentrated in the summer. The Minjiang River, Fujiang River, and Mianyuan River have a strong geomorphological shaping effect, with deep river cuts and well-developed tributaries, and are the main rivers in the study area. The complex geological environment in this area, with the combined effects of external factors such as earthquakes, rainfalls, and human activities, has not only led to the development of geohazards along the main faults and rivers but also formed many distinctive features of geohazards, for example the large quantity, frequent occurrence, zonal distribution, chain reaction, and “hanging-wall” effect [45].

Before the Wenchuan earthquake, the study area had a more significant gradient difference of economic, and the total gross domestic product (GDP) of the western counties was much lower than the average level of Sichuan Province. The Wenchuan earthquake led to a serious decline in economic output in 2008, and the total GDP decreased by about 20% in this year. After the Wenchuan earthquake, the investment of national special prevention and control funds and the inclination of local policies provided effective financial and technical support for regional reconstruction [46]. Since the post-hazard reconstruction, relevant departments had fully considered the regional resource and environmental carrying capacity, and continuously adjusted and optimized the industrial structure based on the scientific determination of the main functions of different regions. Some cities, such as Dujiangyan, Shifang and Mianzhu have become new “growth poles”, relied on tourism, the phosphorus chemical industry and the liquor industry, while the western region has gradually formed a mining industry chain with Wenchuan as the core, and Beichuan has set up a provincial development zone with electronic information and food and drug processing as the leading industries. The total GDP in 2020 was about 4.3 times that in 2008, with an average annual growth rate of about 14%. The rapid economic growth not only effectively alleviated the economic gradient difference between the east and the west but also greatly enhanced the regional ductility and the capability of hazard prevention and mitigation.

### 2.2. Data

#### 2.2.1. Inventory of Geohazards and Nonhazards

The geohazard data was provided by the Sichuan Research Institute of Ecological Restoration of Land Space and Geohazard Prevention and Control and mainly contained information on the type, scale, and location of geohazards after the Wenchuan earthquake. Moreover, this data was mainly oriented to hazard prevention and mitigation, so there were differences in definition compared to those used in remote sensing interpretation or other types of geohazard data. The magnitude of the geohazards in this study was mainly small-scale (78.19%), while middle-scale, big-scale, and huge-scale magnitudes accounted for 18.07%, 2.98%, and 0.76% respectively. We selected 10,325 geohazards by date of occurrence from 12 May 2008 to 20 May 2020, of which 8932 geohazards from 12 May 2008 to 16 October 2017 were used for model construction and 1393 geohazards from 20 March 2018 to 20 May 2020 were used for spatial validation. The geohazards involved in the model construction were dominated by landslides (61.4%), while collapses and debris flows account for 21.4% and 16.8% respectively, which spatially spread along faults and rivers (Figure 2).

The geohazards (positive samples) used for model construction were divided into four periods by taking the years 2008 and 2013 as the time node, while the generation of nonhazards (negative samples) was based on randomness and extensiveness. Combined with the spatial resolution of the data used, the area outside the 2000 m buffer zone of a geohazard was regarded as a “nonhazard zone”, and nonhazard points were randomly generated at an interval of 500 m. Afterward, the missing classes in the categorical factors were added to the nonhazards, and finally, sample points with a positive and negative ratio of about 1:1 were created. In addition, the positive and negative samples were randomly generated into two subsets, including 70% for the training set and 30% for the testing set to facilitate the model construction (Figure A1, Table 1).

#### 2.2.2. Impact Factor of Geohazards

The occurrence of geohazards is restricted by the natural environment and related to human activities, which is often the result of the combined effects of natural and anthropic factors [47]. The complex geological environment superimposed with the impact of external factors, such as earthquake, precipitation, and human activities, can cause a chain reaction to geohazards in the Longmenshan area, which also provides the geological environment basis for constructing the geohazard susceptibility analysis model for our research [48]. Based on full consideration of geomorphology and geological structure of the study area, we selected 14 factors of five types, including topography, geology, land cover, meteorology and hydrology, and human activities, to construct an index evaluation system of geohazard susceptibility by combining the distribution characteristics of geohazards with previous research [49,50,51,52,53]. The data sources of all factors are shown in Table 2, which are consistent with the geohazard data in time and space. Below, the normalized difference vegetation index (NDVI), land use, precipitation, points of interest (POI), and roads are shown for Period I (Figure 3), while the other periods of data appear in Figure A2.

The topographic factor is the key to controlling the potential energy of geohazards, which consists of elevation (Figure 3a), slope (Figure 3b), slope position (Figure 3c) and aspect (Figure 3d). In alpine and gorge regions with frequent earthquakes, the elevation affects the amplification effect of seismic waves and directly affects the potential energy of geohazards [54]. Slope is also an important factor of geohazards, as a steeper slope often represents a higher incidence of geohazards such as landslide and debris flow [55]. Slope position refers to the geomorphic part of the slope surface, which in the river valley is often more prone to geohazards [56]. Aspect determines the hydrothermal conditions and further affects the distribution of vegetation and the weathering characteristics of slope materials, which with poor vegetation and soil stability often has a higher probability of geohazards [57]. Elevation, slope, and aspect were determined using a digital elevation model (DEM).

The geological factor determines the endogenous power and material composition of geohazards, which includes the engineering rock group (ERG) (Figure 3e), weighted Euclidean distance of fault (WEDF) (Figure 3f), earthquake intensity (EI) (Figure 3g), and peak ground acceleration (PGA) (Figure 3h). Engineering rock groups with poor physical and chemical properties will directly affect the stability of the slope, which is an important internal factor to promote the development of geohazards [58]. There are multiple groups of faults with different types and activity intensities in the study area, which can directly or indirectly induce various types of geohazards along the fault through tectonic movement [59]. The severe earthquakes caused by the violent activities of the faults caused different degrees of surface morphology and vegetation destruction within the range of different intensity levels, which in turn affect the number and scale of geohazards [60]. Therefore, EI, which can directly reflect the impact of the Wenchuan earthquake, is taken into account. Over time, the impact of the Wenchuan earthquake has gradually decreased. Therefore, PGA, which reflects the long-term seismic activity, is obviously more representative than EI in periods III and IV [61]. WEDF takes into account the differences in scale, grade, and activity among different types of faults, and was generated by weighted superposition after calculating the normalized Euclidean distance of different levels of faults (main faults were assigned a weight of 1, secondary faults 0.3 and other faults 0.1) [62]. EI and PGA were transformed into continuous data by setting the contour spacing according to the original data.

The land cover factor can reflect the stability of the slope to a certain extent, which is captured by NDVI (Figure 3i) and land use (Figure 3j). Vegetation can stabilize slopes by intercepting surface runoff and slowing soil erosion, which indirectly limits the development and occurrence of geohazards [63]. As an indicator of vegetation coverage and vegetation type, NDVI and land use were considered together [64]. NDVI was represented by annual average values in Period I and Period III and represented by multiyear average values in Period II and Period IV. The land use data used in Period I was obtained by interpreting the remote sensing image after the Wenchuan earthquake, with an additional category of damaged land, which was supplemented with 30 m land use data in 2010 for the missing part, while the land use data in 2012, 2013, and 2017 were used in Period II, Period III, and Period IV, respectively.

Meteorological and hydrological factors affect the driving mechanism and spatial distribution of geohazards and include precipitation (Figure 3k) and Euclidean distance of river (EDR) (Figure 3l). The residual loose materials on the slope by the Wenchuan earthquake and the differences in the topographic gradient make the geohazards restricted by precipitation [65]. Liner erosion and the undercutting of a slope under high flow conditions may cause a slope to become unstable and induce the development of geohazards, and this effect is significantly correlated with the distance from the river [66]. Precipitation was represented by the annual average value in Period I and Period III, and by the multi-year average value in Period II and Period IV. The river network was extracted using a DEM and combined with the spatial distribution of geohazards, the area with a flow greater than 100,000 cfs was retained, and the Euclidean distance was calculated after accuracy verification and qualification according to the existing river system.

The anthropic factor is an exogenous force that affects the geological environment and was represented in our model by POI kernel density (Figure 3m) and the weighted Euclidean distance of road (WEDR) (Figure 3n). POI information usually includes name, category, and coordinates. POI kernel density is often used to identify urban centers, economic vitality, and the intensity of human activities, and the higher its value, the more concentrated human activities are in an area [67]. Compared with nighttime light data, POI kernel density has better temporal-spatial continuity and has been used to characterize human activities in geohazard susceptibility studies [68]. Road construction is representative of human activities, and can include cutting hillsides or changing slope material during construction, which can destabilize slopes and affect the occurrence of geohazards [69]. POI data were imported into ArcGIS 10.2 software for kernel density analysis after crawling and cleaning. WEDR takes into account the differences in the degree of excavation, external load, and spatial distribution of different levels of roads and was generated using weighted superposition after calculating the normalized Euclidean distance of different levels of roads [70]. Due to the limitations on the acquisition of the data, the road data in 2007 were used in period I, the road data in 2012 were used in period II and III, and the road data in 2017 were used in period IV. The road data in 2007 and 2012 included six different levels, national, high-speed, provincial, city and county, township, and others, with weights assigned as 0.85, 0.85, 0.75, 0.4, 0.2 and 0.1, while the road data in 2017 included five different levels, primary, secondary, tertiary, quaternary, extra-grade and others, with weights assigned as 0.85, 0.75, 0.6, 0.4, 0.2 and 0.1, respectively.

We combined the resolution of the impact factors data and the size of the study area. To maintain the consistency of research data, all impact factors were converted into a 90 m × 90 m raster, and a geospatial database was established by combining the geohazard and nonhazard of each period. The coding method of category factors, such as slope position, aspect, ERG, and land use, was to assign integer values starting from 1 to different categories. In addition, in order to reduce the difference between discrete data and continuous data, all impact factors data were standardized.

### 2.3. Method

Our research mainly consisted of the following four steps (Figure 4). First, we prepared the data, including geohazards, non-hazards, and impact factors. Second, we constructed an advanced RF model based on the data and the logical relationship between them, and then output the importance and partial dependence of the factors. Third, we applied accuracy, precision, sensitivity, specificity, recall, the receiver operating characteristic (ROC) curve, and the area under curve (AUC) to verify the accuracy of the RF model. Finally, we predicted the susceptibility of geohazards in each period and analyzed the driving mechanisms of its spatial pattern and dynamics after verifying the reliability of our results. We used ArcGIS 10.2 (Environmental Systems Research Institute, Inc., RedLands, CA, USA), R Studio 3.6 (R Studio, Boston, MA, USA), and other software for data calculation and analysis.

#### 2.3.1. Random Forest (RF)

Random forest (RF) is an integrated algorithm proposed by Breiman to construct multiple decision trees from different data subsets, which can not only solve most classification and regression problems in different fields, but also be used for simulation or prediction [71]. The basic idea of RF is that several independent decision trees make judgments or predictions on the introduced samples and features, then vote on the judgment results of multiple decision trees, and the class that wins the most votes is the model output [72]. RF can handle multidimensional data without feature selection and has a simple implementation process [73], which is superior to traditional statistical methods in geohazard susceptibility studies [74].

The *mtry* and *ntree* are two primary hyperparameters in RF; the former is the number of factors in each decision tree and the latter is the number of decision trees in the model [75]. Usually, the *mtry* is set less than the number of factors and the *ntree* is set 500 to ensure model diversity [76]. The RF model for this study was implemented by calling the *caret* package in R Studio 3.6 software. The model was trained by setting default parameters such as 10-fold cross-validation (to prevent overfitting) and accuracy as the metric, and *ntree* was set to 500 for each period of the RF model because of the max accuracy. Then, the model was trained again by setting *ntree* to 500 and *mtry* from 1 to 13 sequentially (total number of factors), and then the best *mtry* of RF at different periods were determined because of the max accuracy: 9 (I), 3 (II), 10 (III), 4 (IV). Finally, four different RF models were constructed by setting the optimal *ntree* and *mtry* to predict the geohazard susceptibility in four periods.

#### 2.3.2. Partial Dependence Analysis

Partial dependence analysis is based on the machine learning model itself, and its main purpose is to reveal the relative importance of a certain feature or region of the factor to the target [77]. Different features or regions in a factor have different partial effects, and a higher value of effect indicates that the relative importance of the feature or region is greater [78]. The partial dependence plot in this study was implemented by calling the *pdp* package in R Studio 3.6 software.

#### 2.3.3. Evaluation of Model Accuracy

As the output parameters of the model, accuracy, precision, sensitivity, specificity, and recall are usually considered effective indicators to evaluate the accuracy of fitting or prediction (Table 3) [79].

The Kappa index (*κ*) is generally considered more scientific than simple percentage consistent calculation because it takes into account the possibility of coincidence by chance and is therefore used to measure the reliability of the prediction model in this study [80]. The calculation formula is as follows.
(1)$$\kappa =\frac{P_{obs}-P_{exp}}{1-P_{exp}}=\frac{\left(TP+TN/n\right)-\left\{\left(TP+FN\right)\left(TP+FP\right)+\left(FP+TN\right)\left(FN+TN\right)/\sqrt{M}\right\}}{1-\left\{\left(TP+FN\right)\left(TP+FP\right)+\left(FP+TN\right)\left(FN+TN\right)/\sqrt{M}\right\}}$$
where $P_{obs}$ is the observed consistency, $P_{exp}$ is the expected consistency, and n is the proportion of correctly classified samples.

In addition, the ROC curve and AUC value are also commonly used methods for model accuracy evaluation [81]. The ROC curve tests the predictive ability and separability of the model through positive and negative samples, while AUC is the area enclosed by the coordinate axis under the ROC curve. Generally, AUC values range from 0.5 (diagonal) to 1, with values closer to 1 indicating better predictive power of the model. The calculation formula is as follows.
(2)$$\mathrm{AUC}=\sum_{p=1}^{n}\left(X_{p+1}-X_{p}\right)\times \left(S_{p+1}-S_{p}-S_{p}/2\right)$$
where $X_{p}$ is the specificity and $S_{p}$ is sensitivity.

## 3. Results

### 3.1. Model Accuracy Verification

The different evaluation indicators from the confusion matrices of the RF models for the four periods were used to determine the reliability of the prediction results (Table 4). Accuracy and Kappa index were greater than 0.9, which indicated that the RF model performed well in each period. The precision of the RF model for geohazards and nonhazards was greater than 0.9, which indicated that the prediction results were good in all periods. The recall rates of geohazards and nonhazards in the RF model were above 0.95, which verified that the classification outcomes were reasonable in all periods.

In addition, the positive and negative sample points of each period related to the prediction results, and the accuracy of the prediction results in each period was verified by the ROC curve and AUC value. Referring to the previously established classification standard [82], the AUC values were divided into five levels: poor (0.5–0.6), average (0.6–0.7), good (0.7–0.8), very good (0.8–0.9), and excellent (0.9–1). The ROC curve of the RF model in the four periods is shown in Figure 5. The figure shows that the AUC value was between 0.975 and 0.989, all of which were excellent. All RF models had an AUC value of more than 0.9 and excellent performance, thus they can be used in analyzing the geohazard susceptibility of the study area.

### 3.2. Spatial-Temporal Characteristics of Geohazard Susceptibility

The macrodistributional pattern of geohazard susceptibility in the four periods was similar and significantly different between the eastern regions and the western regions (Figure 6). The eastern regions, including Dujiangyan, Pengzhou, Mianzhu, Anzhou, and Jiangyou had high susceptibility indices along the faults; the western regions, including Lixian, Maoxian, Beichuan, and Pingwu had high susceptibility index along the rivers; and Wenchuan and Qingchuan had high susceptibility indices along the faults and rivers.

Combined with the geohazard susceptibility maps, the commonness at all periods and the variation at different periods of the geohazard sensitivity index could be found in space. The average susceptibility index increased from 0.314 in 2008 to 0.353 in 2014–2017 and increased substantially in 2009–2012 and 2014–2017. The change of the average susceptibility index in the 2 km buffer zone of the fault or river can reveal the spatial change of the susceptibility of geohazards at different stages. In 2008, the Yingxiu-Beichuan Fault, the western section of Jiangyou-Dujiangyan Fault, and along the Minjiang, Fujiang, Mianyuan, Tongkou and Baicao rivers were areas that were highly susceptible to geohazards, with an average susceptibility index of approximately 0.65. Compared to 2008, the geohazard susceptibility from 2009 to 2012 changed significantly in the east section of the Jiangyou-Dujiangyan Fault, Pingwu-Qingchuan Fault, and along the Qingjiang, Baicao, and Tongkou rivers. The susceptibility index increased by 0.35, 0.22, and 0.19 in the east section of the Jiangyou-Dujiangyan Fault, Pingwu-Qingchuan Fault, and along the Qingjiang River, respectively, and decreased by 0.20 and 0.13 along the Baicao River and Tongkou River, respectively. Compared to 2009–2012, the geohazard susceptibility in 2013 changed significantly in the east section of Jiangyou-Dujiangyan Fault, and along the Zagunao, Baicao, and Tongkou rivers. The susceptibility index increased by 0.23, 0.22, and 0.18 along the Zagunao River, Baicao River, and Tongkou River, respectively, and decreased by 0.17 in the east section of the Jiangyou-Dujiangyan Fault. Compared with 2013, the geohazard susceptibility from 2014 to 2017 changed significantly along the Jiangyou-Dujiangyan Fault, and along the Baicao, Tongkou, and Kaijiang rivers. The susceptibility index increased by 0.17 in the east section of the Jiangyou-Dujiangyan Fault and decreased by 0.15, 0.13, 0.13 and 0.11 in the west section of the Jiangyou-Dujiangyan Fault, and along the Kaijiang River, Baicao River, and Tongkou River, respectively. In addition, the geohazard susceptibility index along the Mianyuan River was characterized by a slow change and eventually a large decrease of 0.13 in 2014–2017 compared to 2008.

The spatial distribution of geohazard susceptibility during the four periods differed in 20% of the area (Table 5). The correlation coefficient of geohazard susceptibility in the four periods was between 0.798–1, which was characterized by decreases period by period, which indicated that the temporal-spatial difference in geohazard susceptibility was gradually increasing. The correlation coefficient between 2008 and 2013 is the lowest, which indirectly indicated that the Wenchuan earthquake and heavy rainfall led to the partial differences in geohazard susceptibility. The correlation coefficient between 2009–2012 and 2014–2017 was more significant, which indicated that the geohazard susceptibility had more commonalities in the normalization period after major natural disaster events.

The intersection area of the top 20% of the geohazard susceptibility index in the four periods under different combinations accounted for about 23.28% of the total area, and its distribution was around the faults and along the rivers (Figure 7a). The intersection area of the four periods accounted for about 6.51% of the total area, which was concentrated along the Yingxiu-Beichuan Fault and the west section of Jiangyou-Dujiangyan Fault, as well as along the Minjiang, Fujiang, and Mianyuan rivers. The intersection area of the three periods under different combinations accounted for about 8.51% of the total area, which not only had a small distribution along the above the faults and rivers, but also had a distribution mainly concentrated along the Kaijiang River where I, II, and III intersected, the southeast of Jiangyou where I, II, and IV intersected, and the Tongkou, Baicao, and Pingtong rivers where I, III, and IV intersected. The intersection area of the two periods under different combinations accounted for about 8.26% of the total area. In addition to these faults and rivers, the new distribution was mainly concentrated in the eastern section of the Jiangyou-Dujiangyan Fault where II and IV intersected, and along the Mianyuan River where I and II intersected.

Based on the superposition of geohazard susceptibility in the four periods, the distribution of geohazard susceptibility levels in the study area obtained by the equal interval method [83] had a high spatial correlation with the location of geohazards (Figure 7b, Table 6). The areas with high and very high geohazard susceptibility mainly included the Yingxiu-Beichuan Fault, the Pingwu-Qingchuan Fault, and the western section of the Jiangyou-Dujiangyan Fault where tectonic activity is strong, as well as along the Minjiang, Fujiang, Mianyuan, Zagunao, and Tongkou rivers where there was strong gravity unloading, which accounted for 19.5% of the total area, and about 84.22% of the geohazards occurred in these areas. The areas with low and very low geohazard susceptibility mainly included the western mountainous area where frost and weathering are severe and the flat eastern plain, which accounted for 64.62% of the total area and only 2.66% of the geohazards.

### 3.3. Factor Importance and Partial Dependence

As an essential model output parameter, the RF model can estimate the importance of factors by the contribution degree. The larger the importance of the factor, the greater the impact and significance. Due to the different sizes of data sets in different periods, the ranking of factor importance was only considered in the horizontal comparison.

The main driving mechanisms of geohazards in different periods are complex and partially diverse (Figure 8). Period I was primarily driven by topographic, geological, and land cover factors; Period II was primarily driven by topographic, meteorological, and hydrological factors; Period III was driven by topographic, meteorological, and hydrological factors more significantly than Period II; Period IV was mainly driven by topographic, meteorological, hydrological, and anthropic factors. In summary, the impact of the topographic factor was relatively stable and the elevation and slope played an essential role in multiple periods. The impact of the geological factor decreased slowly, and EI/PGA ranked only third in the importance ranking of period I. The impact of land cover had a U-shaped change, especially in period I, when the importance ranking of NDVI was third. The impact of the meteorological and hydrological factors had an inverted U-shape, and the importance ranking of precipitation in period II and period III was first and second, respectively. The impact of the anthropic factors had an upward trend, and the ranking of the importance of the POI kernel density rose to third in period IV.

The partial dependence reflects the overall trend and sensitive range of the continuous factors (Figure 9). The relative importance of elevation rose first and then decreased between 500–3000 m elevation, maintained a high level between 500–1500 m, and gradually stabilized after 3000 m. The relative importance of slope also increased first and then decreased between 5–50°, and maintained a high level between 5–20°, especially in period III. The relative importance of WEDF in the different periods had similar characteristics, which decreased and then increased between 0.9–1.75 and maintained a high level between 1.65–1.75. The relative importance of EI in period I had a prominent peak and a trough in XI and IX intensity regions, respectively, while the trough significantly narrowed and the peak disappeared in period II. The relative importance of PGA in Period III increased and then decreased between 0.1 and 0.2, while in Period IV the relative importance of PGA had a downward trend. The relative importance of NDVI had an apparent trough between 0.77–0.85 in Period I, and had an upward trend and maintained a high level between 0.84–0.92 in Period III. The relative importance of precipitation in Period I and Period III decreased at first and then rose between 800–970 mm and 960–1250 mm, respectively, and maintained a high level between 900–970 mm and 1170–1250 mm, respectively. The relative importance of precipitation in Period II and Period IV rose at first and then decreased between 800–1050 mm, and maintained a high level between 820–930 mm. The relative importance of EDR continued to decline between 0–3000 m, had a high level between 0–2000 m, and gradually stabilized after 3000 m. The relative importance of POI kernel density, which represented the intensity of anthropic activity in different periods, rose from 0–10 to the peak and then dropped to a stable level, and the value of the POI kernel density at the peak gradually increased. The relative importance of WEDR had a downward trend in different periods, and fluctuated substantially between 0–0.7 in Period I and Period III.

The partial dependence revealed the categories that were sensitive and the impact differences in different categories (Figure 10). The relative importance of slope position increased as the position of the slope in the longitudinal profile decreased, which was the highest in the valleys and the lowest at the ridges and upper slopes. For the relative importance of aspect, north, northeast, and east were greater than south, southwest, and west in Period I, but the opposite was found in other periods. The relative importance of ERG in Period I and III was the highest in clastic rock intercalated carbonate rock and metamorphic rock intercalated carbonate rock, respectively, and did not show apparent characteristics in other rock groups. The relative importance of land use was generally high in crop land, low in forest, and fluctuated greatly in urban areas and grassland.

## 4. Discussion

### 4.1. Verification of Geohazard Susceptibility

Both new geohazards and previous studies were used to verify the reliability of prediction results. From 20 March 2018 to 20 May 2020, we collected 1393 geohazards, which were dominated by landslides (1101). Overlaying the new geohazards with Figure 7b, we found that about 69.13% of the geohazards were in very high or high susceptibility areas, while only 0.57% of the geohazards were in very low susceptibility areas. The spatial correlation between new geohazards and the distribution of geohazard susceptibility in the study area indicated that our predictions were reliable. Also, the distribution of geohazard susceptibility was highly consistent with previous studies, except that the very high or highly susceptibility areas were less frequent along the Pingwu-Qingchuan Fault [84]. Combined with the geohazard susceptibility in different periods (Figure 6), we found that although the geohazard susceptibility along the Pingwu-Qingchuan Fault gradually increased, there was still a significant gap with the other three main faults. From a regional perspective, our distribution of geohazard susceptibility agreed well with those of previous studies in the main counties (cities) such as Wenchuan [85], Beichuan [86] and Qingchuan [87].

Our results indicated that the random forest model can be used to investigate geohazard susceptibility in our study area. Previous studies also proved the applicability of the model in the Longmen mountain region after the Wenchuan earthquake [39]. In addition, some researchers used the random forest model to quickly map the occurrence probability of secondary geohazards in a short time after the earthquake and their results were highly accurate [88]. Some researchers used this model and combined it with satellite images for the long-term monitoring of geohazards [89]. Thus, the random forest model has good applicability for geohazard identification, monitoring, and assessment.

### 4.2. Analysis of Spatial-Temporal Characteristics of Geohazard Susceptibility

We found that the susceptibility of geohazards from 2008 to 2017 gradually increased, which indicated that the Longmen Mountain region remained highly susceptible to geohazards ten years after the Wenchuan earthquake. Our finding agrees with previous views [90]. The Yingxiu-Beichuan Fault, the seismogenic fault of the Wenchuan earthquake, maintained a high geohazard susceptibility pattern in the ten years after the Wenchuan earthquake, which was similar to the western section of the Jiangyou-Dujiangyan Fault, and along the Minjiang and Fujiang rivers [91]. The dynamic changes of geohazard susceptibility mainly occurred along the Pingwu-Qingchuan Fault, the eastern section of the Jiangyou-Dujiangyan Fault, and along the Mianyuan River and many secondary rivers. Due to the activation of the seismogenic fault, the geohazard susceptibility along the Pingwu-Qingchuan Fault increased substantially after 2008, and remained highly susceptible in the later periods [39]. The geohazard susceptibility in the eastern section of the Jiangyou-Dujiangyan Fault was significantly higher in 2009–2012 and 2014–2017, which, combined with the importance ranking and partial dependence, maybe caused by a combination of multiple factors, such as many low mountains and hills, complex geological structures, intensive cropland use, and precipitation in the area [92]. Affected by heavy precipitation, the geohazard susceptibility along the secondary rivers, such as the Zagunao, Tongkou, Baicao, Pingtong, and Qingjiang rivers in 2013 exceeded that in 2008, and reached the highest level in the ten years after the Wenchuan earthquake [93]. Because of the gradual reduction of loose materials and the slow vegetation recovery after the Wenchuan earthquake, the geohazard susceptibility along the spatially adjacent Mianyuan and Kaijiang rivers gradually declined [94].

### 4.3. Cause Analysis of Spatial-Temporal Variation of Geohazard Susceptibility

Combining the importance ranking of the factors (Figure 8) and partial dependence (Figure 9 and Figure 10), our study comprehensively analyzed the spatial distribution, dynamic changes, and causes of geohazard susceptibility.

#### 4.3.1. Topographic Factors

Topography plays a significant role in controlling geohazards and largely determines the spatial distribution of geohazard susceptibility. In our study area, there were many high mountains and valleys with large elevation differences and the undercutting of hillsides by river erosion, which form a large number of high and steep slopes that provide the preconditions for geohazards [95]. As one of the main controlling factors, elevation in this region had a significant gradient in the range of 500–3000 m and was related to the valley morphology. The elevation at around 1500 m corresponds to the area where the river enters the canyon from a wide valley, with steep terrain, long-term gravity unloading, prominent seismic response, relatively fragmented rock, and the highest frequency of geohazards [96]. The river valley above 1500 m is relatively flat and the area above 3000 m is controlled by strong frost and weathering, thus these areas are less susceptible to geohazards due to more gentle slopes and infrequent human activity [97].

As one of the main control factors, slope plays a decisive role in controlling surface water runoff and the accumulation of loose materials on the slope [98]. Slopes of 5–50° are the most susceptible to geohazards in the study area, and geohazards on slopes of 5–20° are mainly controlled by the superposition of faults, lithology, rivers, and other factors [99], and the development of geohazards on slopes in the range of 20–50° is based on sufficient effective slip spaces and loose deposits [100]. Different slope positions have various topographic and geomorphic characteristics, which result in differences in overlying rock and gravity potential energy [101]. Valleys are widely distributed along rivers, with deep cutting and strong weathering of rock, which provide the conditions conducive to geohazards [102].

Slope aspect has different characteristics under different hydrothermal conditions and the direction of seismic wave propagation, which indirectly affects the development of geohazards [103]. The study area is located in the northern hemisphere, and the south, southwest, west, and northwest aspects have longer exposure to sunlight, lower soil moisture, and better vegetation cover, while the northeast, east, north, and southeast aspects have shorter exposure to sunlight, higher soil moisture, lower vegetation cover, and poorer soil stability [104]. The southeast aspect was most susceptible to geohazards in 2008, was perpendicular to the main fault direction, and reflected the “fault misdirection effect” of geohazard occurrence [105]. The north, northeast, and east aspects were the dominant aspects in other periods, and vegetation and soil stability on these slopes had an indirect impact on the occurrence and development of geohazards [106].

#### 4.3.2. Geological Factors

As the constituent materials of geohazards, the composition and physicochemical property of the engineering rock group have significant impacts on the occurrence of geohazards. In 2008 and 2013, clastic rock intercalated carbonate rock and metamorphic rock intercalated carbonate rock had a significant impact on geohazards in the study area. These two rock groups are mainly composed of slate, sandstone, quartz sandstone, and carbonate rock, which have a weak resistance to weathering and erosion, poor mechanical properties, such as permeability and shear strength [107], and are prone to collapse, landslides, and provide abundant material for debris flow [108]. In 2009–2012 and 2014–2017, geohazards ceased in all rock groups, and no individual rock group had a significantly higher probability of occurrence than other groups. In our four periods, the importance ranking of ERG fluctuated in the middle midstream, which may be caused by the impact of lithology that was partially reflected in factors such as slope and the presence of a river [109]. In addition, referring to the entire earthquake area, other related studies also indicated that the influence of lithology is not very clear, and topographic factors have the most significant control on the distribution of geohazards [110].

Fault affects landform, rock mass structure, and slope stress field and stability, which has the significant characteristics of activity difference and distance attenuation in the four periods. Frequent geological tectonic movement in the study area leads to broken rock formations, developed joints, strong weathering, and large reserves of loose materials along faults, which provides the dynamic conditions and material for geohazards [111]. Combined with the sensitive range of WEDF in partial dependence, this effect is more significant within 10 km of faults, which limits the extension range of geohazard susceptibility along the main faults to a certain extent. In different periods, the Wenchuan earthquake eruption brought about short-term tectonic activity differences between the main faults [112], and the activation of the other faults by a seismogenic fault during energy transfer caused the difference in tectonic activity between different faults to decrease gradually [113]. The impact of precipitation rose after 2008 and further strengthened in 2013, and thus the impact of faults has experienced a process of rising, falling, and rising again in the periods after 2008.

An earthquake caused by violent fault activity could directly induce geohazards. For example, the Wenchuan earthquake originated from the Yingxiu-Beichuan Fault, which caused many slopes to become unstable along the fault and on both sides of the river valley perpendicular to the fault by destroying mountain and vegetation, and induced a large number of geohazards [114]. Therefore, EI, as a characterization of the seismogenic effect of the Wenchuan earthquake, had a significant impact in 2008. The nonunique distribution characteristics of geohazards led to the partial dependence of EI in 2008 generating a significant peak and trough in XI and IX intensity areas, respectively. After the Wenchuan earthquake, the activity difference between the main faults decreased, and the loose deposits left by earthquakes in the mountains triggered geohazards that were mainly caused by precipitation [115]. Therefore, in 2009–2012, the ranking of EI decreased significantly, and relative importance in different intensity areas also decreased significantly. The PGA, which characterizes the long-time series seismic activity in the study area, was in the middle of the factor importance ranking, and its partial dependence was characterized by the inverted U-shape in 2013 and the fluctuation decline in 2014–2017. One possible reason was that PGA in the study area changed significantly in the direction perpendicular to the seismogenic fault, while geohazards were distributed along the faults and the rivers perpendicular to the faults, which resulted in the correlation between geohazards and PGA being affected by the distribution characteristics of geohazards within different PGA ranges [61]. The distribution of geohazards along both sides of the river increased substantially, especially in 2013, which was consistent with the change direction of PGA, and led to an upward trend in the partial dependence of PGA between 0.10–0.13.

#### 4.3.3. Land Cover Factors

The impact of NDVI on geohazards and susceptibility distribution was more significant in 2008 and decreased in the other periods. The high vegetation cover in the study area and the geohazard was mainly controlled by faults and topographic factors, and thus the impact of NDVI on geohazards was significant only in a specific range [116]. Vegetation damage caused by the Wenchuan earthquake and its secondary geohazards was concentrated on the banks of rivers and in areas with serious surface rupture [117], which resulted in apparent differences in NDVI in the surrounding surfaces in 2008. In 2008, the partial dependence of NDVI was evident in the range of 0.77–0.85, which was distributed in the southeastern plains, most of the northern region, and the areas along the rivers (about 3 km), where nonhazards were more widely distributed. In 2013, the spatial distribution of NDVI in the range of 0.84–0.92 covered multiple secondary rivers, such as the Zagunao, Tongkou, and Pingtong rivers, and correlated with the distribution of geohazards in the same year, so the partial dependence of this range was high. However, the short-term heavy precipitation in 2013 weakened the ability of vegetation to suppress geohazards [118], and resulted in a decrease in the importance ranking of NDVI, although we found a high level of partial dependence.

Different types of land have different soil consolidation and slope protection capabilities, and respond differently to external factors, such as earthquakes, precipitation, and human activities, which lead to significant differences in their impact on geohazards. The forests in the north and west of the study area were widely distributed, with good vertical structure and deep and wide roots, which can effectively intercept precipitation and slowly transfer water through the canopy and understory vegetation layer, the litter layer, and the root soil layer. Thus, forest is less susceptible to geohazards [119]. Cropland was mainly distributed in the southeastern plains and western valleys, which were strongly affected by earthquakes, had weak soil consolidation and shear strength, and had cultivation, and were thus more susceptible to geohazards [120]. The land use data for 2008 classified the urban areas and grasslands damaged by the Wenchuan earthquake as damaged land [121], which caused a weaker susceptibility of urban land and grass land to geohazards during this period relative to other periods.

#### 4.3.4. Meteorological and Hydrological Factors

Precipitation had two different impact mechanisms in the different periods, a superposition effect with the earthquake and an amplification effect on topographic factors, which are another main reason for the changes in geohazard susceptibility. After the Wenchuan earthquake, the deformation and rupture of the mountain created a large amount of loose material with a general degree of consolidation, a large void ratio, and poor internal cohesion in the slope [122], which affected the hazard threshold of precipitation and resulted in the geohazard in 2009–2012 to be mostly caused by precipitation and its erosion of loose deposits [115]. In 2013, continuous heavy precipitation broke through the regional hazard threshold, changed the surface water runoff, the slope deadweight pore static pressure, the shear strength of the weak interlayer, the hydrostatic and hydrodynamic pressure, and induced a large number of landslides and further evolved into debris flows [123]. The areas sensitive to precipitation in all periods corresponded to valleys and faults where geohazards were concentrated, especially the expansion of sensitive areas in 2013.

The influence of rivers had significant distance attenuation characteristics, which limited the spatial distribution of geohazard susceptibility. The water system in the study area was widely distributed, and the rivers had a strong downward cutting effect, forming a series of V-shaped valleys. The undercut erosion and lateral erosion of the river erodes the river bank, which makes the slope steeper, increases the slip spaces, weakens the stability of the slope, and superimposes external force factors such as earthquakes and rainfall, which can easily induce geohazards [124]. The magnitude of this influence is related to the proximity of the river [125]. The relative importance of EDR in our study was more significant within 2000 m from the river, and tended to be stable when it was farther than 3000 m.

#### 4.3.5. Anthropic Factors

The impact of POI kernel density has multiple patterns of action and is mainly concentrated along the river valleys in the west and the hilly, piedmont areas in the east. The complex geological environment and large river drop in the study area provide humans with abundant mineral and hydropower resources, and also limit human construction projects [126]. Unreasonable hydropower projects, mining, and village and road construction along the western river valley have caused serious vegetation damage and the large accumulation of solid materials, which has exposed rocks and intensified weathering, reduced the carrying capacity of the geological environment, and provided conditions conducive to geohazards [127]. The establishment of a large number of croplands in the eastern piedmont area has caused local soil erosion and river uplifting or silting, and has deteriorated the ecological and geological environment [128]. In addition, the indirect effects of human activities are amplified in the years when strong earthquakes and heavy rainfall occur, which result in a large number of houses, roads, and farmlands being threatened by geohazards [129].

Restricted by topography, most of the main roads in the study area are distributed along both sides of the river valley, which has impacted geohazards that are closely related to the geomorphic features and geological environment along the roads [130]. The spatial distribution of the WEDR sensitive range in partial dependence covers the main national and provincial roads in the study area, such as G213, G317, S302 and S105. The G213 and G317 roads in the area run along the Minjiang River and Zagunao River, respectively, where high and steep slopes and a large number of exposed areas of bedrock lead to strong unloading and rock weathering. Provincial roads, such as S302 and S105, approach and cross the Yingxiu-Beichuan Fault and Pingwu-Qingchuan Fault. Frequent fault activities have caused the shallow rock mass of the slope along the line to loosen and strongly deform, and the source of the loose materials has increased. The artificial effects, such as slope cutting and top-loading, during road construction were more prominent before the Wenchuan earthquake and were amplified by the Wenchuan earthquake, which induced many geohazards [131]. Therefore, to effectively reduce the susceptibility of geohazards along the road during the reconstruction of the road after the Wenchuan earthquake, a variety of technical methods, such as partial evasion, tunnel crossing, and stone walls were adopted to help alleviate the geohazard risk [132].

### 4.4. Implications and Limitations

The Wenchuan earthquake triggered a series of surface processes that formed a chain of geohazards [133], and its damage to the stability of the mountains and vegetation also affected the development of subsequent geohazards for a long time, which seriously threatened the development of the regional economy [134]. Here, we used different time periods to research the spatiotemporal changes in geohazard susceptibility, the results of which provide a scientific basis for hazard prevention and control under different situations and to help the effective development of the areas hit hard by the Wenchuan earthquake. With the gradual weakening of the Wenchuan earthquake impact and the change of the primary driving mechanisms, geohazards, such as landslides and debris flows, were still highly active in the decade after the Wenchuan earthquake. In the subsequent research and management process, researchers should pay more attention to the superposition effect of earthquake and precipitation as well as the amplification effect of precipitation on topographical factors, and actively adjust and optimize regional hazard prevention. In the future, we should focus on prevention and control in the areas that are very highly and highly susceptible to geohazards, such as the areas along the main faults, and along the Minjiang and Fujiang rivers. In accordance with scientific investigation and a monitoring system, an emergency response system should be guaranteed, and relocation avoidance or engineering measures should be taken if necessary. In high and medium geohazard susceptibility areas, such as along the Zagunao, Tongkou, and Qingjiang rivers, effective detection and early warning should be a priority and reasonable safety distance and protective facilities should be set up, especially in the rainy season from July to September every year. In addition, the construction of towns and roads along the high mountains valleys in the west should be based on scientific planning and prevention and control measures must be utilized to effectively control the negative impact of human activities on the regional geological environment and reduce the risk of geohazards.

There were some limitations or shortcomings in our study. In terms of theory, our study had a limited portrayal of hazard-causing mechanisms, and thus the selection of impact factors might not be comprehensive. Also, the influence of the lag effect of some factors was not considered. The rupture and deformation of mountains caused by tectonic movement, the infiltration and erosion of precipitation on slopes, and the response of engineering rock groups to inducing factors needs to develop to a certain extent before they lead to geohazards [135]. The destruction of slope structure caused by human engineering activities, the stability difference caused by the influence of hydrothermal conditions in different aspects, and the difference of runoff interception ability of different vegetation takes a longer time to affect the process of geohazards from conception to occurrence. Under different conditions, the times before collapse and landslides are different, even on sunny days, which is the objective manifestation of the lag effect of such geohazards [136]. In terms of data, because the management department is mainly oriented toward hazard prevention and mitigation, there were differences in the definitions used for the interpretation of geohazard data obtained from remote sensing and field investigation. The type, stability, and scale of geohazards were not specifically considered in this study. In addition, the limited number of sample points might reduce the RF model’s recognition of categorical factors. All impact factors had two types of continuous data and categorical data, and had differences in data accuracy and data integrity, which might have affected our results. Machine learning algorithms, such as random forest, are more tolerant of continuous data than categorical data, which does not affect the prediction results but impacts the factor importance. Due to the spatial scale and differentiation effect, the RF model predictions in some regions might be weakened. A more refined study can be carried out in the future according to the type and scale of geohazards and combine other models or algorithms to synthesize the impact of different factors on geohazards.

## 5. Conclusions

A time series study of geohazard susceptibility can not only enhance the reliability of previous studies that studied a single time period, but also accurately identify geohazard formation mechanisms, and be an effective tool for preventing and assessing regional geohazards. Here, we selected 12 counties (cities) in the Longmen Mountain region and 14 factors in five categories, topography, geology, land cover, meteorology and hydrology, and anthropology, to construct a geohazard susceptibility index evaluation system. We used an advanced RF model to study the changes and driving mechanisms of geohazard susceptibility during the Wenchuan earthquake and the subsequent decade. Our study was divided into four periods based on the Wenchuan earthquake and heavy precipitation: 2008, 2009–2012, 2013, and 2014–2017.

We had four main conclusions. First, the Longmen Mountain region remained in the highly active stage of geohazards from 2008 to 2017. While geohazard susceptibility gradually increased, there was a significant correlation between the main faults and rivers, which was consistent with the findings of previous studies.

Second, the geohazard susceptibility along the Yingxiu-Beichuan Fault, the western section of the Jiangyou-Dujiangyan Fault, and along the Minjiang and Fujiang rivers remained relatively stable and high. The geohazard susceptibility along the Pingwu-Qingchuan Fault increased significantly after 2008, the geohazard susceptibility on both sides of the Jiangyou-Dujiangyan Fault had a “low-high” temporal variation, the geohazard susceptibility along the Mianyuan and Kaijiang rivers gradually decreased, and geohazard susceptibility along the secondary rivers, such as the Zagunao River, Tongkou River, and Baicao River peaked in 2013.

Third, the impact of topographic factors was relatively stable. The impact of the geological factors decreased after 2008 while the impact of the meteorological and hydrological factors increased significantly. The impact of the land cover factors had a U-shaped change, and the impact of the anthropic factors had an upward trend.

Fourth, elevations in the range of 500–3000 m and 5–50° slopes had a significant effect on geohazards, and largely determined the spatial distribution of geohazard susceptibility. The impact of the Wenchuan earthquake gradually weakened and was accompanied by the superposition effect of precipitation on earthquakes and the amplification effect on topographic factors in the post-2008 period. In addition, the main control ranges of faults and rivers were 10 km and 3 km, respectively. The impact of the POI kernel density had many forms and was concentrated along the western river valley and the eastern piedmont area; the impacts of roads were limited by the geomorphological features and geological environment along the roads.

## Figures and Tables

**Figure 1 ijerph-19-03229-f001:**
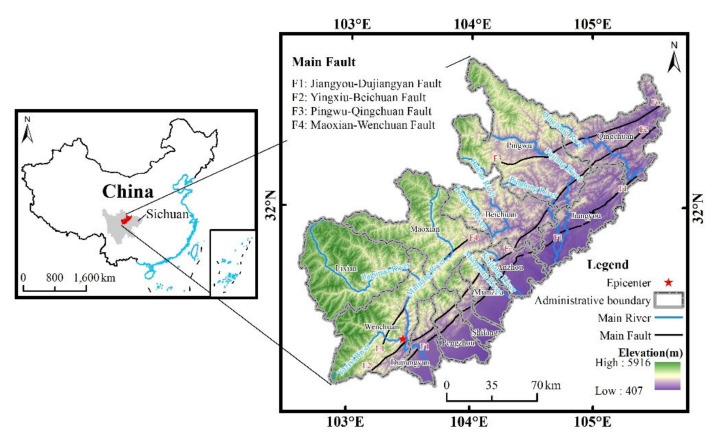
Location of the study area.

**Figure 2 ijerph-19-03229-f002:**
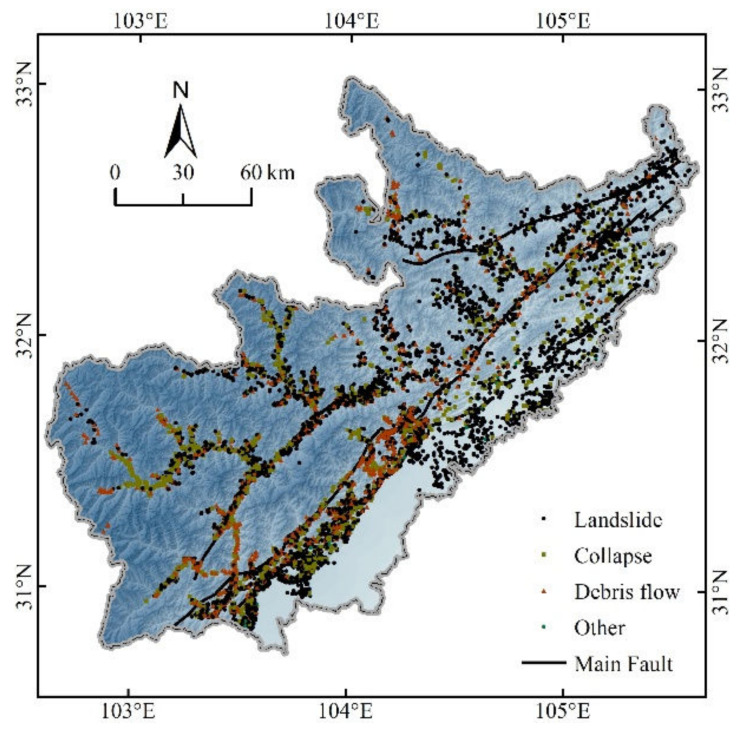
Geohazard inventory map of the study area.

**Figure 3 ijerph-19-03229-f003:**
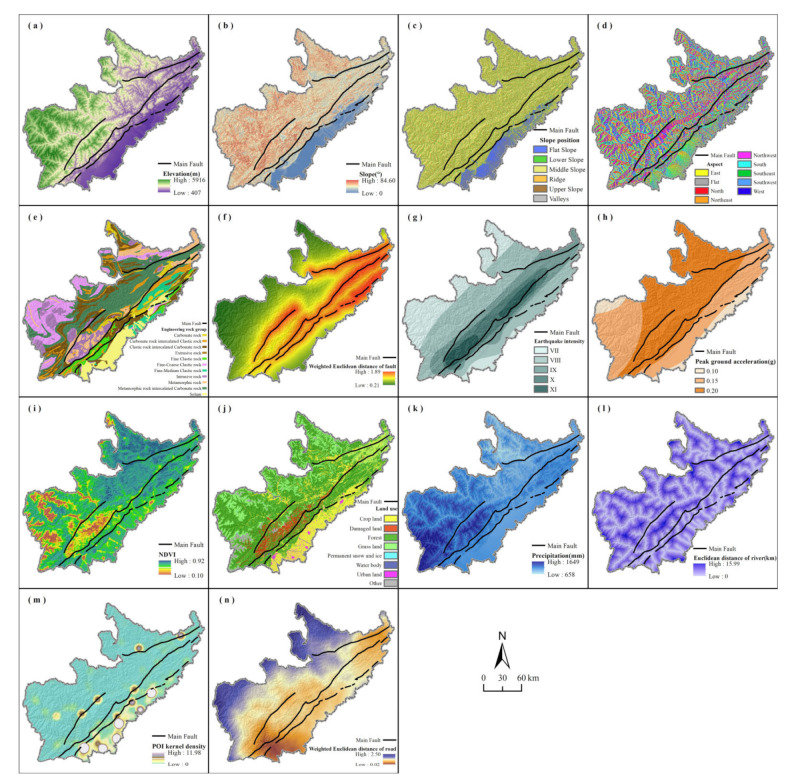
Thematic maps of impact factors: (**a**) elevation; (**b**) slope; (**c**) slope position; (**d**) aspect; (**e**) engineering rock group; (**f**) weighted Euclidean distance of fault; (**g**) earthquake intensity; (**h**) peak ground acceleration; (**i**) NDVI; (**j**) land use; (**k**) precipitation; (**l**) Euclidean distance of river; (**m**) POI kernel density; and (**n**) weighted Euclidean distance of road.

**Figure 4 ijerph-19-03229-f004:**
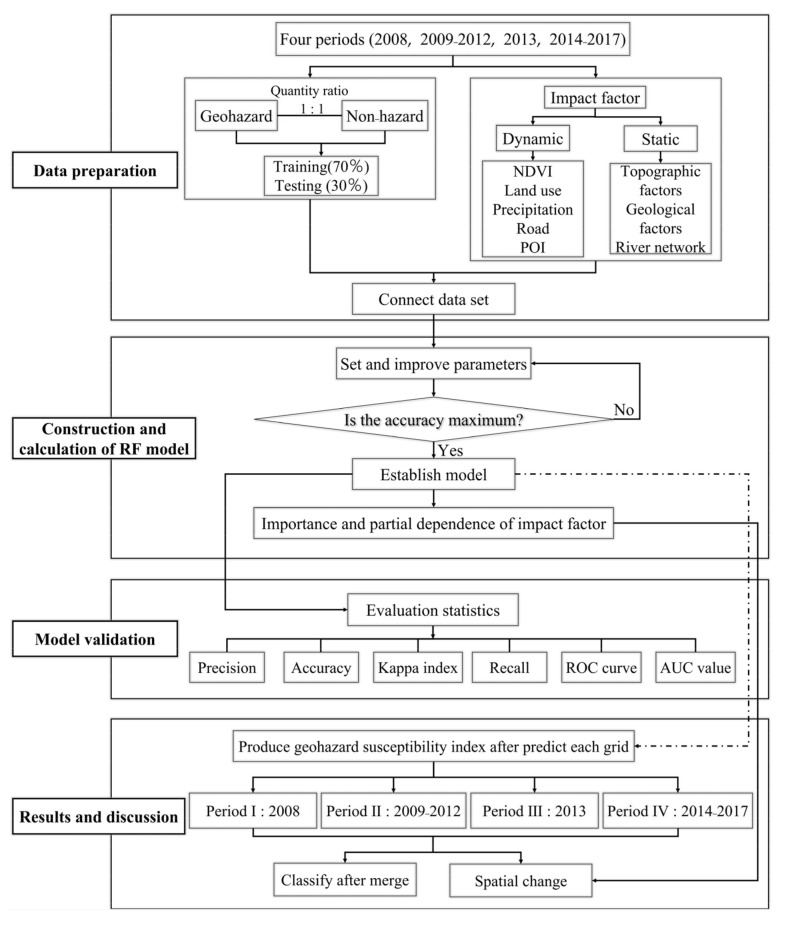
Study flow chart.

**Figure 5 ijerph-19-03229-f005:**
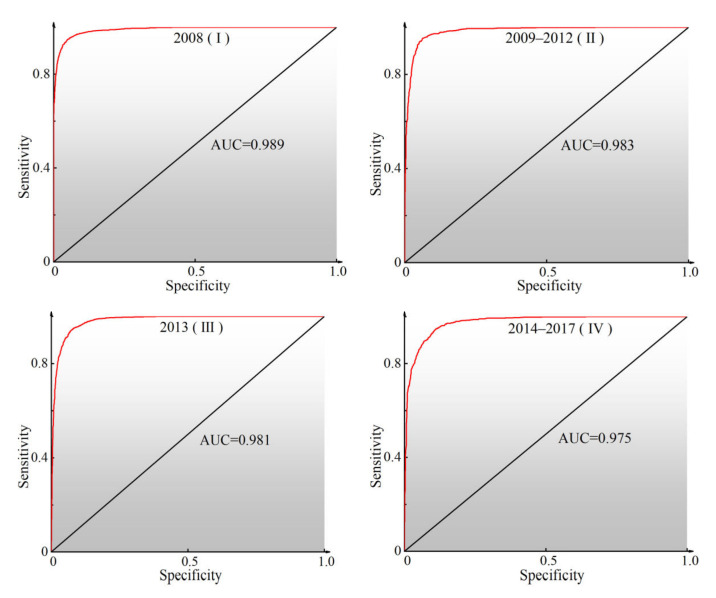
ROC curve and AUC value in four periods.

**Figure 6 ijerph-19-03229-f006:**
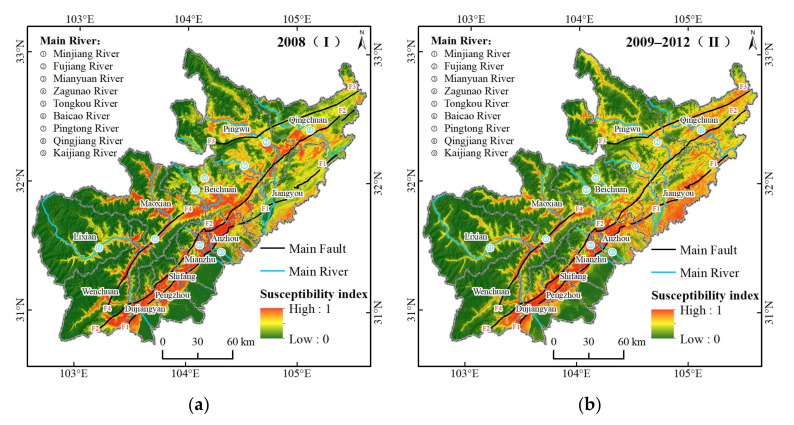
Geohazard susceptibility in four periods: (**a**) 2008 (I); (**b**) 2009–2012 (II); (**c**) 2013 (III); and (**d**) 2014–2017 (IV).

**Figure 7 ijerph-19-03229-f007:**
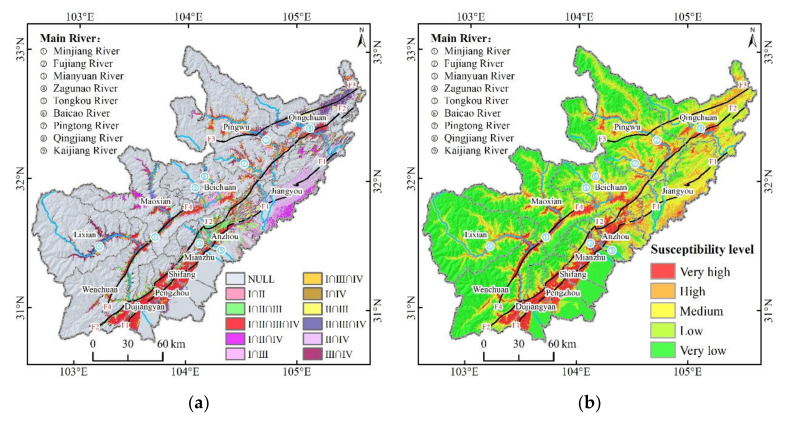
Synthesis of four periods of geohazard susceptibility: (**a**) high susceptibility intersection diagram in four periods (I is 2008; II is 2009–2012; III is 2013; IV is 2014–2017); and (**b**) geohazard susceptibility in 2008–2017.

**Figure 8 ijerph-19-03229-f008:**
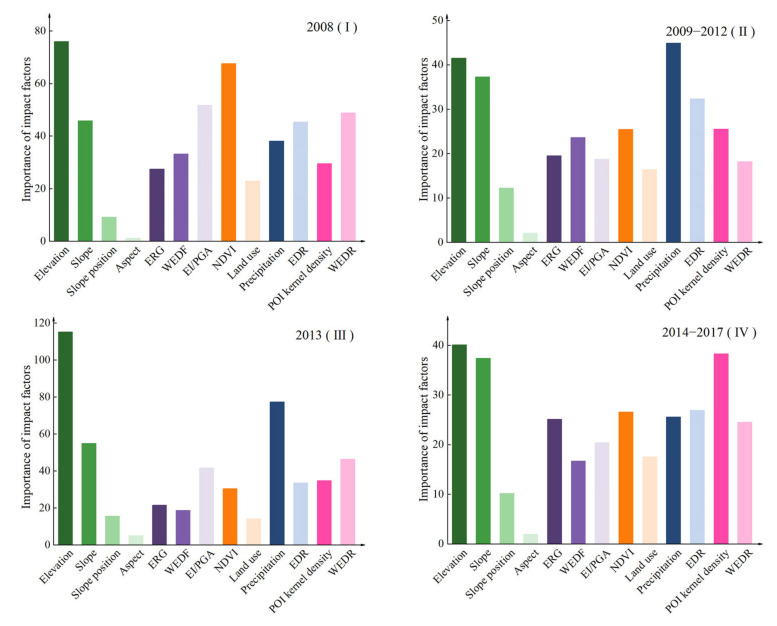
Importance of impact factors in four periods.

**Figure 9 ijerph-19-03229-f009:**
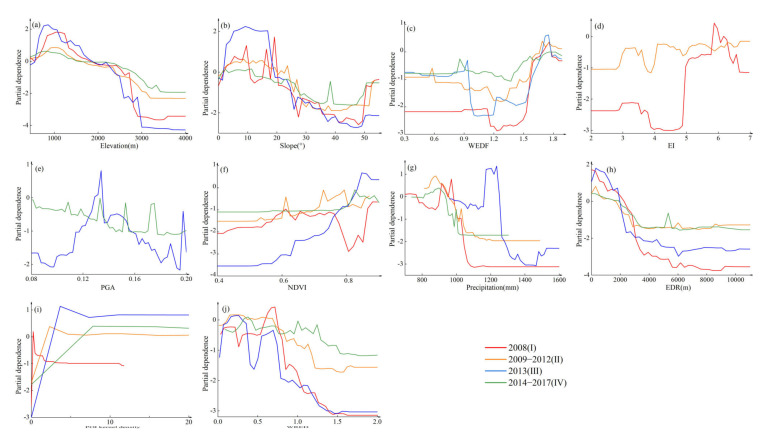
Partial dependence plot in continuous factor: (**a**) elevation; (**b**) slope; (**c**) WEDF; (**d**) EI (from 2 to 7 is intensity VII to intensity XI); (**e**) PGA; (**f**) NDVI; (**g**) precipitation; (**h**) EDR; (**i**) POI kernel density; and (**j**) WEDR.

**Figure 10 ijerph-19-03229-f010:**
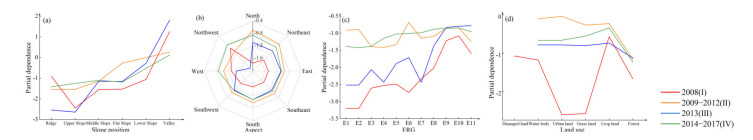
Partial dependence plot in category factor: (**a**) slope position; (**b**) aspect; (**c**) ERG (E1 is extrusive rock; E2 is solum; E3 is intrusive rock; E4 is carbonate rock; E5 is fine-coarse clastic rock; E6 is fine-medium clastic rock; E7 is carbonate rock intercalated clastic rock; E8 is metamorphic rock; E9 is fine clastic rock; E10 is clastic rock intercalated carbonate rock; and E11 is metamorphic rock intercalated carbonate rock); and (**d**) land use.

**Table 1 ijerph-19-03229-t001:** Number of geohazard and nonhazard points in four periods.

Period	Year	Geohazard		Nonhazard	
		Training	Testing	Training	Testing
I	2008	2213	949	2215	949
II	2009–2012	1004	430	1008	433
III	2013	2120	909	2120	909
IV	2014–2017	915	392	916	392

**Table 2 ijerph-19-03229-t002:** Data and data sources.

Type	Data Name	Data Sources	Spatial Resolution
Topographic factor	Elevation	ASF Data Search (https://search.asf.alaska.edu/) (Accessed on 20 May 2021)	15 m×15 m
	Slope	ASF Data Search (https://search.asf.alaska.edu/) (Accessed on 20 May 2021)	15 m×15 m
	Slope position	Geospatial Data Cloud (https://www.gscloud.cn/) (Accessed on 20 May 2021)	90 m×90 m
	Aspect	ASF Data Search (https://search.asf.alaska.edu/) (Accessed on 20 May 2021)	15 m×15 m
Geological factor	Engineering rock group	China Geological Survey (https://www.cgs.gov.cn/) (Accessed on 22 May 2021)	1:500,000
	Fault	China Geological Survey (https://www.cgs.gov.cn/) (Accessed on 22 May 2021)	1:500,000
	Wenchuan earthquake intensity	China Earthquake Administration (https://www.cea.gov.cn/) (Accessed on 23 May 2021)	Vector Data
	Peak ground acceleration	China Earthquake Administration (https://www.cea.gov.cn/) (Accessed on 23 May 2021)	Vector Data
Land cover factor	NDVI	NASA (https://modis.gsfc.nasa.gov/) (Accessed on 24 May 2021)	1000 m×1000 m
	Land use	ESA (https://maps.elie.ucl.ac.be/CCI/viewer/) (Accessed on 25 May 2021); GLOBELAND30 (https://www.globallandcover.com/) (Accessed on 25 May 2021)	300 m×300 m 30 m×30 m
Meteorological and hydrological factor	Precipitation	Resource and Environment Science and Data Center (https://www.resdc.cn/) (Accessed on 25 May 2021)	1000 m×1000 m
	River network	NASA (https://www.nasa.gov/) (Accessed on 28 May 2021)	Vector Data
Anthropic factor	POI	Gaode Open Platform (https://lbs.amap.com/) (Accessed on 1 Jul 2021)	Vector Data
	Road	National Catalogue Service for Geographic Information (https://www.webmap.cn/) (Accessed on 3 Jul 2021); NAVIINFO (https://www.navinfo.com/) (Accessed on 5 Jul 2021)	Vector Data

**Table 3 ijerph-19-03229-t003:** Formula and definition of evaluation parameters.

Metric	Equation	Definition
ACC	ACC=TP+TN/M	The proportion of geohazards and nonhazards points which are correctly classified
Precision	Precision=TP/TP+FP	The fraction of relevant instances in the retrieved instances
SST	SST =TP/TP+FN	The percentage of geohazards points that are correctly classified
SPF	SPF=TN/TN+FP	The percentage of nonhazards points that are correctly classified
Recall	Recall=TP/TP+FN	The proportion of positive samples predicted to be correct

*TP* is the number of correctly predicted geohazard points; *FP* is the sum of nonhazard points classified as geohazard points; *TN* is the number of correctly predicted nonhazard points; *FN* is the sum of geohazard points classified as nonhazard points; *M* is the sum of geohazard points and nonhazard points.

**Table 4 ijerph-19-03229-t004:** Confusion matrix of different RF models.

Period	Year	Prediction	Reference		Summation	Kappa
			Geohazard	Non-Hazard		
I	2008	Geohazard	3084	75	Precision: 0.976	0.952
		Nonhazard	78	3089	Precision: 0.975	
		Summation	Recall: 0.975	Recall: 0.976	Accuracy: 0.976	
II	2009–2012	Geohazard	1402	51	Precision: 0.965	0.942
		Nonhazard	32	1390	Precision: 0.977	
		Summation	Recall: 0.978	Recall: 0.965	Accuracy: 0.971	
III	2013	Geohazard	2954	62	Precision: 0.979	0.955
		Nonhazard	75	2967	Precision: 0.975	
		Summation	Recall: 0.975	Recall: 0.980	Accuracy: 0.977	
IV	2014–2017	Geohazard	1276	54	Precision: 0.959	0.935
		Nonhazard	31	1254	Precision: 0.976	
		Summation	Recall: 0.976	Recall: 0.959	Accuracy: 0.968	

**Table 5 ijerph-19-03229-t005:** Correlation of four periods of geohazard susceptibility.

Period (Year)	I (2008)	II (2009–2012)	III (2013)	IV (2014–2017)
I (2008)	1.000	0.833	0.824	0.798
II (2009–2012)	0.833	1.000	0.848	0.881
III (2013)	0.824	0.848	1.000	0.826
IV (2014–2017)	0.798	0.881	0.826	1.000

*p* value less than 0.05

**Table 6 ijerph-19-03229-t006:** Geohazard susceptibility number and proportion for different classes of RF.

Geohazard Probability	Susceptibility Level	Grid Number	Area Proportion (%)	Geohazard Number	Geohazard Proportion (%)	Density Proportion (Pcs/km^2^)
<0.55	Very low	1,733,079	42.37	13	0.14	0.001
0.55–1.90	Low	910,353	22.25	225	2.52	0.031
1.90–2.75	Medium	649,682	15.88	1172	13.12	0.223
2.75–3.30	High	464,832	11.36	2515	28.16	0.668
>3.30	Very high	332,727	8.14	5007	56.06	1.858

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
