# Peer review of "Susceptibility Analysis of Geohazards in the Longmen Mountain Region after the Wenchuan Earthquake"

_ijerph, 2022, doi:10.3390/ijerph19063229_

Round 1

Reviewer 1 Report

The authors have constructed a multi-hazard map for Longmen Mountain Region after Wenchuan Earthquake. Following suggestions can be incorporated in the manuscript to enhance its quality.

Abstract: 1. Authors can mention the geohazards that they have considered in the study. 

Introduction: 2. BY geohazard, do authors refer to any particular geohazard like earthquake triggered landslide? This must be made clear in the introduction. Because abstract gives a picture like multi-hazard susceptibility is mapped. 

3. page 3, line 102 - what dynamic changes are considered in the study

4. The summary of work presented at the end of the introduction must be improved considerably to highlight the unique contribution of the work and its application

Objectives and Overall approach 5. Objective can be part of the introduction and overall approach can be included in materials & methods

6. 3.2.1 6. Can authors include the magnitude of the landslides, debris, etc. The different types of landslides (geohazards) must be described in the present geoenvironmental setting

7.  3.2.2 p.6, line 203 - expand POI, lines 207-208, how do authors relate elevation with factors like seismic waves, roads, etc.

8. Authors must describe each factor with respect to specific categories in a factor where geohazard/landslides are abundant

9. p.7, lines 235-236 - must justify with suitable references

10. Authors must include details of precipitation, like the annual rainfall, rainfall months, variability, intensity over the study area. What attribute of precipitation was considered for modelling?

11. section 3.2.2 must be improved significantly for better understanding of the reader

12. Why 90 m x 90 m resolution was selected?

13. Quality of figures 4, 5 and 6 must be improved

14. Factors causing landslides, debris flow and other geohazards are not the same. How do authors account for using the same factors to model all geohazards in the region.

15. section 4.2 - authors must clearly indicate the advantage of combing the spatial susceptibility maps, lines 427-428, p. 15

16. Quality of figures 8 , 9 and 10 must be improved

17. section 4.2 and 5.2 have to be clearly distinct

18. General: The paper must be structured for better readership

19. Conclusion can be brief highlighting the merits of the study and its application in the selected area. 

20. Authors must take care to avoid repetition of content in the various sections. 

Author Response

Dear Reviewer:

We’re very grateful for your comments and suggestions, which are quite valuable and important for improving the quality of our manuscript. Based on the comment and request, we have made extensive modification on the original manuscript.The following is my reply to your comments.

Reviewer 2 Report

I want to start by thanking you for the opportunity to review “Susceptibility Analysis of Geohazards in the Longmen Mountain Region after the Wenchuan Earthquake," it was an interesting project that looked at the possible influences on geohazards and how technology, modeling, and machine learning can be utilized to asses the risks associated from these hazards.

Overall, I thought this was a fine manuscript and would be be appropriate for publication in its present form. I think the study and results were clearly presented and it was easy to follow and understand the goal of the project. If only to have a suggestion I would recommend the authors expanding on the applicability of this process and speaking to its reproducibility, if only in terms of how these methods are applied. There is a brief sentence or two on utilizing this in some other specific areas, but what is the applicability in other regions or more general settings?

I wish the authors well in their pursuit of research on this topic, and I appreciated the chance to review this work.

Author Response

(The authors gave the same response as above.)

Reviewer 3 Report

The article suggests an interesting and attractive topic. The effort made is evident, possible questions are answered in section 5.4 Implications and Limitations.   I attach only one  comment:

Line 108 -  abbrevation RF is not explained  

 I  recommend the manuscript to be published

Author Response

(The authors gave the same response as above.)
